# Real-World evidence revelations: The potential of patient support programmes to provide data on medication usage

Eszter Palffy[1]*, David John Lewis[2,3]

1 Novartis Ireland Limited, Dublin, Ireland, 2 Patient Safety & Pharmacovigilance, Development, Novartis Pharma GmbH, Wehr, Germany, 3 School of Life and Medical Sciences, University of Hertfordshire, Hatfield, England

☉ These authors contributed equally to this work.
* eszter.palffy@novartis.com

**Data Availability Statement:** The minimal data set that was used for the research is provided as a Supporting Information file. Further data is output from the Novartis database and access will be granted on reasonable request. The best point of

## Abstract

Patient Support Programmes (PSPs) are used by the pharmaceutical industry to provide education and support to consumers to overcome the challenges they face managing their condition and treatment. Whilst there is an increasing number of PSPs, limited information is available on whether these programmes contribute to safety signals. PSPs do not have a scientific hypothesis, nor are they governed by a protocol. However, by their nature, PSPs inevitably generate adverse event (AE) reports. The main goal of the research was to gather all Novartis-initiated PSPs for sacubitril/valsartan, followed by research in the company safety database to identify all AE reports emanating from these PSPs. Core data sheets (CDS) were reviewed to assess if these PSPs contributed to any new, regulatory-authority approved, validated signals. Overall, AEs entered into the safety database from PSPs confirmed no contribution to CDS updates. Detailed review of real-world data revealed tablet splitting or taking one higher dose tablet a day instead of twice daily. This research, and subsequent analyses, revealed that PSPs did not impact safety label changes for sacubitril/valsartan. It revealed an important finding concerning drug utilisation i.e. splitting of sacubitril/valsartan tablets to reduce cost. This finding suggests that PSPs may contribute important real-world data on patterns of medication usage. There remains a paucity of literature available on this topic, hence further research is required to assess if it would be worth designing PSPs for collecting data on drug utilisation and (lack of) efficacy. Such information from PSPs could be important for all stakeholders.

## Introduction

PSPs are not designed to investigate a scientific hypothesis nor, typically, are they governed by a protocol, hence these programmes are not safety-oriented. However, by their nature, these programmes generate potentially reportable suspected Adverse Drug Reactions (ADRs). Additionally, patients too can report AEs via these programmes that may contain valuable information.

contact is the QPPV group mailbox required under European Law as the central contact point for safety enquiries which is constantly monitored 24/7/365. eu-eea.qp@novartis.com.

**Funding:** The author(s) received no specific funding for this work.

**Competing interests:** The authors have declared that no competing interests exist.

Current European legislation, *Directive 2010/84/EU and GVP Module VI (2017)* requires all *suspected adverse reactions* including those from PSPs to be collected and recorded by the Marketing Authorisation Holder (MAH) [1, 2]. The question arises whether this high-volume of cases from PSPs leads to any validated signals that further strengthen existing knowledge of the safety profile of the medications involved. The current system for European Pharmacovigilance (PV) is summarised in a system manual. The regulatory framework was recast in June 2012 and the new documentation comprised both legislation (e.g. European Commission's Implementing Regulation 520/2012) and guidelines (EMA's Good Vigilance Practice (GVP) modules) [3–5]. Prior to the commissioning of this framework, earlier regulations, directives and guidelines did not provide precise instructions for all areas of PV systems, particularly market research (MR) and PSPs. The lack of detailed or definitive guidance led to a wide range of operational standards, which varied, from MAH to MAH.

Since 2012, the pharmacovigilance and governance of PSPs have evolved. However, the underlying question, whether such efforts from the MAH lead to any newly identified safety signals related to the medicinal product, is still not clear. Three major literature reports gave strong background for this research; the work of Portnoff et al. in 2017, describing the result of a survey of MAHs in Europe and two research papers published by Jokinen et al in 2019 as part of the TransCelerate initiative to explore the value of safety information sources [6–8]. However, neither of the papers analysed if the efforts by the MAH in collecting and reporting AEs from PSPs add any value to the extant signal detection process. Most MAHs have a system in place to ensure the appropriate management and oversight of PSPs. This limited evidence in the public domain supports the need for more research to determine if the 2012 PV legislation had positive impact on patient safety in this respect. Indeed, there is a bigger question concerning whether there have been any tangible public health benefits from the significant investment of time and resources applied by MAHs to govern and manage PSPs.

In the past 20 years, PSPs have evolved from a primarily call centre setting for reimbursement to a patient-centric service to support adherence, disease management, and provide financial assistance. The main goal is often to allow the MAH to stay close to their customers.

In the US, MAHs fund different assistance programs to support patients in reducing or eliminating out-of-pocket payments and reducing paperwork associated with health insurance schemes [9]. As opposed to the US, in Ireland, most innovative MAHs operate according to the applicable Code of Practice for the Pharmaceutical Industry (IPHA Code), published by the Irish Pharmaceutical Healthcare Association (**"IPHA"**). The IPHA Code outlines that a PSP must have the objectives of monitoring disease activity, achieving better healthcare outcomes, enhancing patient care, must be non-promotional, must not be designed as an inducement to prescribe or operated in a promotional manner and does not offset the routine business practice costs of the recipient [10].

## Company definitions

At the company, the term *patient-oriented programmes (POPs)* have been coined and adopted for all activities where the MAH interacts in a post-marketing organised data collection system without a protocol with healthcare professionals, patients and caregivers. Within POPs, there is a subgroup dedicated to *patient support programmes (PSPs)*. The company definition of PSPs was created due to lack of guidance in requirements in EMA GVP Annex 1 Definitions (Rev 4) that contained no guidance on what constituted a PSP and did not provide a definition of MR. Included in the general POP classification at the time of research, there were *non-PSP programmes*, such as Market Research, Disease Management Programmes, and other programmes that are classified within the POP definition, but not falling within the PSP category.

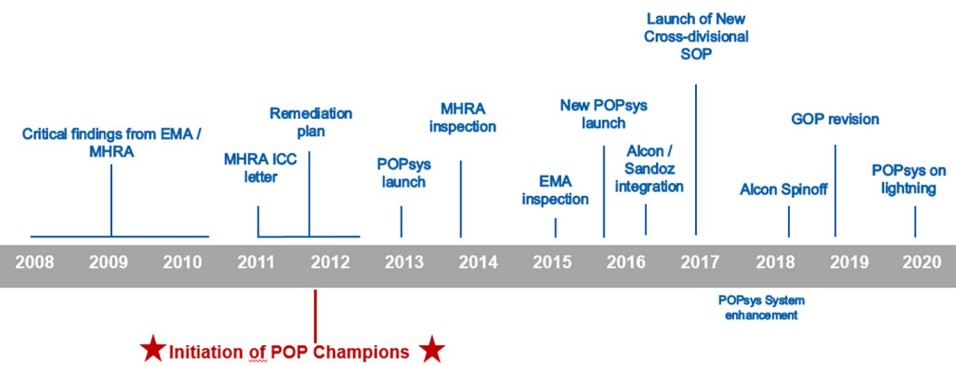

**Fig 1. Inspection findings highlighting process gaps and POPsys implementation history [11].**

This group was formed to provide a uniform governance standard for organised data collection systems, as defined within GVP VI. Within the non-PSP classification, the company created two groups as Non-PSP High with high probability and Non-PSP Low with low probability of receiving an AE.

## The evolution of PSPs

PSPs have been through multiple iterative changes that have shaped the processes that constitute current best practices within the company. The below timeline (Fig 1) outlines the evolution of PSPs at the company.

Following the critical inspection finding on PSPs by MHRA the "Novartis POP Remediation Project" was commenced in 2011. By 2013, the MHRA re-inspection yielded no further critical findings and confirmed the issues with PSPs resolved. As recommended by Portnoff et al 2017, a repository was essential to oversee PSPs and their relevant information therefore the company introduced a system, called POPsys to fulfil it [6]. POPsys is validated as per the Code of Federal Regulations Part 11 and GxP compliant system for tracking POPs within the company. It is used for documenting appropriate quality standards at specific milestones as well as assessing, tracking and approving External Service Providers (ESP). Additionally, POPsys is used for tracking the pharmacovigilance monitoring activities i.e. reconciliation and source data verification (SDV) for applicable programmes.

The initially introduced POPsys was updated multiple times to accommodate the growing workload due to the integration of other acquired companies under the same platform, fulfil regulatory and audit requirements, ensure a stable platform and environment. Since this research, the company introduced yet another upgrade of the system in November 2020 to align it with its updated processes. During this process change, the former classification of PSP and non-PSP has been changed along with the POP definition.

## Materials

### About the product

The sacubitril/valsartan combination was first approved and registered in the United States of America (USA) on 7 July 2015 (International Birth Date—IBD), followed by the EMA approval on 25 September 2015. Novartis is currently the MAH in 111 countries worldwide. Sacubitril/valsartan is registered for the treatment of symptomatic chronic heart failure with reduced ejection fraction (HFrEF) in adult patients. The formulation is a film-coated tablet contained two components, valsartan and sacubitril supplied in different strengths combined

in one tablet i.e. 24mg/26mg (50mg), 49mg/51mg (100mg) and 97mg/103mg (200mg). The usual starting dose is 24mg/26mg twice daily which is increased to the target maintenance dose of 97mg/103mg after 2 to 4 weeks, as tolerated.

### Identified PSPs and cases

All AEs from the 64 sacubitril/valsartan PSPs were extracted from the company global safety database ARGUS on 09 October 2020. The search identified 32,334 ICSRs. No cases or events were excluded from the listing. MedDRA version 23.1 was used for all analyses. All events in the cases were presented verbatim with their corresponding MedDRA terms.

### Company core data sheet and signal management

As per GVP Annex I–Definitions, the core data sheet is a document prepared by the MAH containing in addition to safety information, materials relating to indications, dosing, pharmacology and other product information. All relevant safety information that is contained in the CDS must be listed in all countries as a minimum requirement. There are exceptions, but only when the national or regional regulatory authority requires a specific modification [12].

At Novartis, there are two types of changes to the Core Data Sheet (CDS), where the CDS either shows a full version update e.g. 2.0 or a minor version update e.g. 1.1. The rationale behind these changes is that while the major versions (e.g. 1.0, 2.0) are CDS creations and updates, the minor versions (e.g. 1.1, 2.5) are CDS amendments. The CDS update at Novartis is a pre-planned periodic review (e.g. every 3, 5 or 7 years) depending on the complexity of the safety profile of the products, as assessed by the PV department. In contrast, CDS amendments as a result of validated signals are entirely reactive in nature; a standardised process to implement variations is used to amend the labels in between creation and the next scheduled update.

For the purpose of this research, all of the documented CDS changes (updates and amendments) were reviewed for sacubitril/valsartan in order to identify ADRs added to the CDS. The review focused on five safety-relevant sections of the CDS, namely:

5.–Contraindications
6.—Warnings and precautions
7.—Adverse drug reactions
8.–Interactions
9.—Pregnancy, lactation, females and males of reproductive potential
10.–Overdosage

The reason for selecting these sections was that they are most relevant to pharmacovigilance; and data collected from ICSRs has the propensity to impact on the content of the aforementioned sections of the CDS.

### Method

The research was conducted in POPsys, to gather all POPs for sacubitril/valsartan created by the company. Following the confirmation of PSPs for the selected innovative product, research was commissioned in ARGUS to identify all adverse event reports emanating from these PSPs. The extracted data were analysed to identify signals raised and validated from the cohort of sacubitril/valsartan PSPs by reviewing changes of the CDS from the first version (CDS at launch) to the current one. Additionally, the line listing was analysed to review the acceptance and adherence information from a patient perspective.

The research study did not require ethical approval. All PSP participants consented to participate in the patient support programmes and consented to adverse event reporting. The

**Table 1. Number of POPs per classification.**

| Non-PSP High | Non-PSP Low | Out of scope | PSP | TBC | Grand Total |
|---|---|---|---|---|---|
| 6,973 | 3,873 | 603 | 1,589 | 439 | 13,477 |

data used for the research was presented as cohort line listing with no identification of individual patients during the research; data privacy and security was maintained throughout the study. The datasets generated and analysed during the review are available from the corresponding author on reasonable request.

## Results

As of 29 April 2020, there were 13,477 programmes recorded in POPsys globally. This number included all classifications in the system i.e. PSP, Non-PSP High, Non-PSP Low, programmes out of scope and programmes where classification is to be confirmed (TBC). Table 1 shows the numbers per classification.

Table 2 showed that there were 1,589 PSPs in the company POPsys database globally as of 29-Apr-2020 in different stages. Out of 1,589 PSPs, there were 64 PSPs on sacubitril/valsartan.

A review of the sacubitril/valsartan CDS showed two newly identified ADRs i.e. "hypersensitivity (including rash, pruritus, and anaphylaxis)" and "dizziness" added to the core label on 10 July 2017 and 24 June 2020 respectively.

Out of the 32,334 cases from the PSPs, the sacubitril/valsartan listing revealed a high number of ICSRs reporting hypersensitivity reactions (including rash, pruritus, and anaphylaxis) and dizziness. However, after causality assessment, only nine ADRs were identified that were consistent with sacubitril/valsartan-associated hypersensitivity; in contrast, there were 583 causally-associated reports of "dizziness". Table 3 summarises the results.

**Table 2. Number of PSPs per programme status.**

| Program Status | Number of PSPs | Comments |
|---|---|---|
| Classification approved | 46 | Programmes are still in a set-up stage and not active or recruiting |
| Classification initiated | 8 | |
| Classification rejected | 5 | |
| Classification review pending | 2 | |
| Initiation approved | 25 | |
| Initiation initiated | 8 | |
| Initiation rejected | 3 | |
| Initiation review pending | 2 | |
| Monitoring ongoing | 471 | Active programmes or to be closed following completion of outstanding PV activities |
| Close-out monitoring ongoing | 28 | |
| Closed | 746 | Closed programmes |
| Cancelled | 228 | Programmes terminated or cancelled for various reasons e.g. erroneous/ duplicate entry, change of local strategy, de-prioritisation, no patient recruitment, classification changed or health authority approval (e.g. Turkey) to run a PSP was not granted. |
| Terminated | 17 | |
| Grand Total | **1,589** | |

**Table 3. Summary of sacubitril/valsartan AE line listing.**

| Event | AEs reported from PSPs | Suspected ADRs | Serious ADRs | Non-serious ADRs |
|---|---|---|---|---|
| Hypersensitivity (including rash, pruritus, and anaphylaxis) | 387 | 9 | 7 | 2 |
| Dizziness | 2,379 | 583 | 100 | 483 |

Hypersensitivity (including rash, pruritus, and anaphylaxis) was discussed and endorsed at the Medical Safety Review Board (MSRB) while, dizziness was added to the label based on the fact that this is a non-serious low impact risk.

Additionally, the review and analysis of the sacubitril/valsartan AE listing revealed that in 9,665 ICSRs from PSPs, the report referred to "*use of a split tablet*" or administration "*once daily use instead of twice daily*". The focus countries were India (4776), Mexico (1809) and USA (1304) with high number of such reports and Hungary (1) and United Kingdom (33) with low number and Ireland with no reports on the "*use of a split tablet*" or "*once daily use instead of twice daily*".

## Discussion

Due to the nature of certain PSPs and the possibility of frequent interactions with patients, the chance of receiving AEs may be high. The collection of AEs is an important element of pharmacovigilance that contributes to signal detection and risk minimisation. In order to identify new safety signals and conduct correct risk minimisation activities, data must be of good quality and must be collected using the most appropriate methods to allow proper assessment of the safety signals. Fig 2 shows that PSPs generate high volumes of cases, but their credibility of the evidence is the lowest. Additionally, there is limited literature available to date that assessed the contribution of PSPs to newly identified signals.

There was one important finding from the sacubitril/valsartan AE listing that was not a safety signal, but highly relevant to PV. This finding was that patients were either splitting tablets or took one daily dose (one higher strength tablet) instead of taking sacubitril/valsartan twice daily (at the recommended strength). This finding may also reveal a pattern of off-label

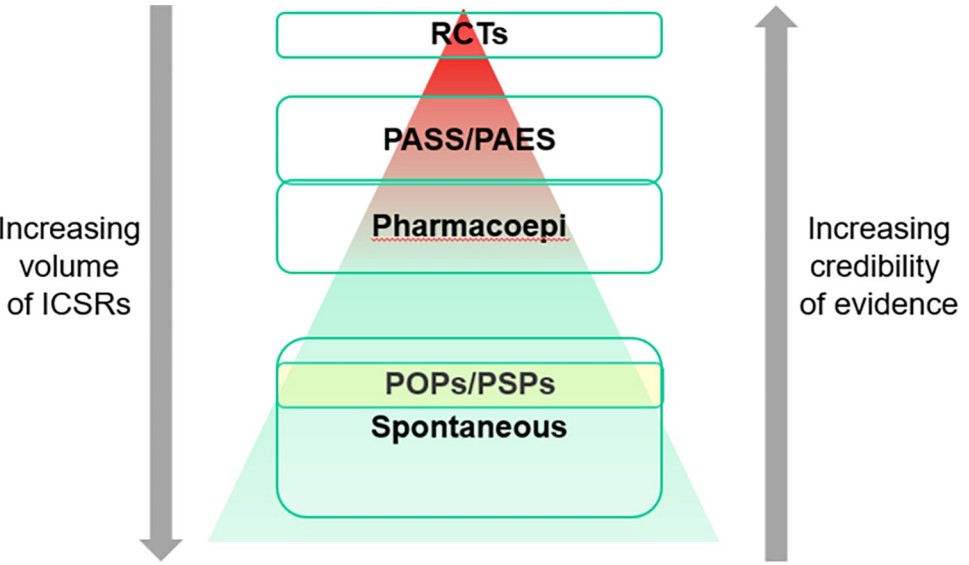

**Fig 2. ICSR credibility hierarchy vs case volume [6].**

use according to the EMA definition that "*refers to any intentional use of an authorised product not covered by the terms of its marketing authorisation and therefore not in accordance with the SmPC*".

This research reviewed PSPs for a company product and assessed whether these contributed to newly identified ADRs that were added to the core label. While PSPs remain an important part of the pharmaceutical business supporting patients and caregivers, this research concluded that PSPs did not contribute to the inclusion of new ADRs on the sacubitril/valsartan label. Notably, the study conducted by Jokinen et al included close to 1,200,000 cases in total from PSPs, Market Research Programmes (MRPs) and social media sources over three years and yielded nine valid signals but none was a medically confirmed high-impact signal leading to a label change or update to the CDS. Only four signals out of nine were identified from PSPs.

As the results showed, sacubitril/valsartan had two label changes, one in 2017 to include hypersensitivity including rash, pruritus, and anaphylaxis and one in 2020 to include dizziness, however, neither of the ADRs added to the core labels were supported by the sacubitril/valsartan PSPs.

Neither of the changes observed in the documented safety profile of sacubitril/valsartan occurred as a result of the safety data generated from PSPs. Signal detection and signal management was conducted in real time and on an ongoing basis, using disproportionality statistics calculated according to the method detailed by Berlin et al. [13]. Observations of disproportional statistics undergo medical and scientific review according to the signal management process in the diagram below (Fig 3):

In addition to routine use of the disproportionality tool, we review designated medical events

(DME) and assess signals of increased frequency, noting trends over time increased severity measured by proxy as increased fatal or hospitalisations. At no stage did any of the safety data from PSPs lead to a change in the documented safety profile of sacubitril/valsartan. Consequently, there were no changes to the core safety information for the product, and no safety variations were submitted after a thorough evaluation of the safety data from the PSPs.

The above findings from the company PSPs have further supported the 2019 Jokinen et al. paper where the research revealed no signals from PSPs lead to a label change or update to the CDS [7, 8].

There were 9,665 reports of patients splitting the sacubitril/valsartan tablet or taking one higher strength dose daily. These off-label uses were either initiated based on instructions from an HCP, or instituted by the patient. There were three countries (USA, Mexico and

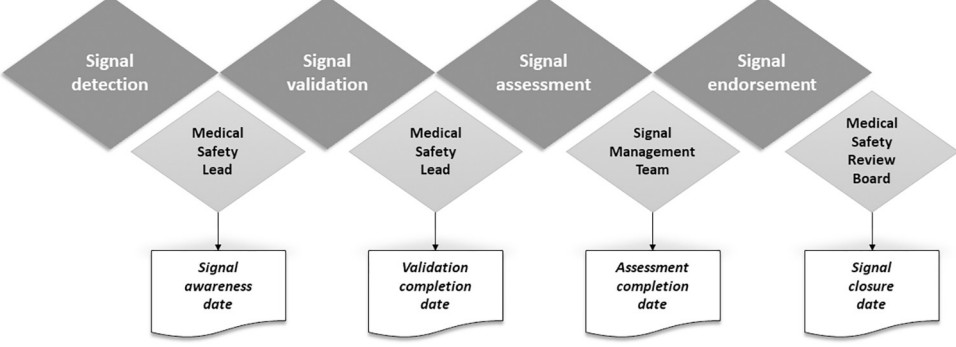

**Fig 3. Signal management process.**

India) with a broad uptake of sacubitril/valsartan where a high number of reported cases of tablet splitting or using a once-daily dose. In some of these cases, this practice was reported due to financial constraints. In contrast, in two countries (Hungary and United Kingdom) there were relatively very low numbers of such cases. Similarly, Ireland had no such events reported despite there being a PSP for sacubitril/valsartan running in the country. The next section looked into whether the low number of such events could be because the majority of prescriptions were paid for by the national health services in the different counties, and not the patient. Therefore, patients were not contributing large amounts of cash to the payment for the prescription as opposed to the US, Mexico and India.

The financial element of "saving" money on medication may be relevant in the three countries with relatively high numbers of reports i.e. India, Mexico and US and available literature supported this theory detailed in the below section. In these countries, patients pay for their medication mainly out-of-pocket despite available governmental support and the cost of medicine is relatively high and not well regulated [14–16].

In India, easy access to various medications and poor community literacy, especially in rural areas, may lead to potential misuse and/or overuse of medicinal products. Additionally, according to data from WHO from 2016, an average of 65% of healthcare expenses is out-of-pocket expenditures (OOPE) versus the world average of ca. 20%. However, affordable medication that is an important part of successful treatment can be enforced by price regulation. In India, the Ministry of Chemicals and Fertilisers via Drug Price Control Orders (DPCOs) by the National Pharmaceutical Pricing Authority achieved more regulated pricing by increasing the regulated medicines from just 74 between 1995 and 2012 to 347 post 2013 [14]. Similarly, in Mexico, there is an increased spend on medicinal products, with limited price control. There is the same problem as in India that medicinal products, especially innovative or patented products, are financed mainly out of pocket. Additionally, the patent creates another layer of problems, which is a lack of competition. This leads to a major challenge to define a fair price and establishing a price control system [15].

On the other hand, in the US, the government does not directly regulate drug prices, but companies are allowed to set their prices. The US system operates largely on private insurance schemes where discounts are negotiated by third parties for private payers or patients may use Medicare, a public health program for the poor. In the US, patients directly pay for about 14% of prescription medicine costs out of their own pockets, which seems a reasonable share however due to high prices this can consist of a significant contribution from patients that are not always affordable [16].

These three examples may explain why the sacubitril/valsartan line listing showed the practice of splitting tablets or taking one tablet daily. This practice was not only patient invented to reduce cost but also prescribed or recommended by HCPs.

"Pill-splitting" is a common practice especially in the US to reduce the cost of the medicinal product. This practice means cutting larger-dose tablets in half to double the amount of smaller-dose pills. Therefore, pill-splitting could lead to lower overall cost. According to Choe et al, 2007, 59% of patients are willing to do so to save on their prescription costs in comparison to 12% who would not do so. This is not a new initiative and many studies available in the public domain as well as physicians, health insurers, and pharmacists recommend it. This trend could result in significant cost savings without compromising efficacy or safety, but only when careful controls are implemented. It is also important to emphasise that not every pill can be split. Only certain types of pills, such as cholesterol control, blood pressure and antidepressant tablets, are recommended for splitting. Additionally, patients should have their physician's consent for doing this. However, this recommendation is not necessarily well known among HCPs and patients [17].

Tablet splitting in the US is common and certain health insurers (e.g. Anthem, Premera, and United Healthcare) introduced a "Half Tablet Program" and only the above-mentioned medications are included. Additionally, the US FDA published a "Best Practices for Tablet Splitting" guide in 2013 due to this recommendation by healthcare companies [18].

The sacubitril/valsartan AE listing clearly showed that this was already a practice, even though this neither supported by the company nor recommended in the package insert. Additionally, the sacubitril/valsartan tablets were not scored but three different strengths available made splitting tablets unnecessary. The incorrect practice of cutting this medicine e.g. uneven halves or wastage of powder and fragments can pose a risk to patient health or may lead to potential lack of efficacy of the medicinal product due to the reduced dose of the active ingredient. In addition, this practice might be particularly dangerous for heart failure patients where miscalculation can lead to an adverse or potentially fatal reaction.

Other counties were reviewed, where these events were reported in significantly lower numbers i.e. UK and Hungary. It is also noted that there was no such practice reported from Ireland. In Ireland, the main perception support scheme was called Drugs Payment Scheme (DPS) which applies to any ordinarily resident in the country defined by the Health Service Executive (HSE) as a "*person who has been living or intends to live in Ireland for a minimum of one year*". Under the DPS, a maximum of €80 in a calendar month was paid by the patient or family registered under the patient. The scheme covers the person, spouse/partner and children aged under 18 (or under 23 if in full-time education). To follow the monthly spending, it would be advised to use the same pharmacy to avoid paying more than monthly cap. If a person paid over the maximum monthly amount, a refund can be requested. Although it must be taken into consideration that €80 per month for a single professional who was generally healthy, could be a significant amount and most likely rarely reached [19].

As opposed to the Irish system, the UK NHS represents a simpler system for patients, which is rather more cost-effective from a patient's perspective. The prescription fee in the UK is £9.15 per item. Prescription prepayment certificates (PPCs) offer savings. This allows patients to pay £29.65 who would normally fill a prescription for four or more items in three months and £105.90 for more than 12 items in one year.

Additionally, prescriptions are free for patients in the following categories:

- Over the age of 60, under 16 or between 16 and 18 if in full-time education

- Pregnant women or had a baby in the previous 12 months

- Patients with specified medical conditions or certain physical disability

- War pensioners

- NHS inpatients

- Patients who have a valid NHS certificate for full help with health costs [20].

In comparison to both UK and Ireland, the Hungarian system provides subsidies to any Hungarian nationals and/or citizens with valid residency who are employed. The employer is mandated to pay the compulsory national insurance for each employee. Those unemployed may decide to pay this fee themselves to keep the governmental support on prescription medication and GP visits. In Hungary, GP visits are free of charge as in the UK while prescription medications are subsidised. Health insurance or "*társadalombiztosítás (TB)*" covers medicines, infant formulas, healing services such as spa treatments, medical swimming, medical gas bath; medical aids and their repair or rental fee is also included in the support i.e. subsidised price. This medicinal product or service is prescribed by the GP while the price is shared between the

patient and the national health insurance agency "*Nemzeti Egészségbiztosítási Alapkezelő (NEAK)*". The patient is only required to pay the subsidised price of the medicinal product and/or service [21].

Ireland, Hungary and the UK have very different healthcare systems, and thus there are different approaches to support patients that require prescription medicines; in all three countries, sacubitril/valsartan tablet splitting was not common. From a patient's perspective, this reflects positively on the European system of pricing regulation when compared to the US or Mexico. Practically in Europe, governments negotiate directly with the MAH to limit what the state-funded health systems pay. For example, countries may refuse to pay for medication that is widely used in the US for pharmacoeconomic reasons e.g. there is less value for money or where a cheaper medication from the same drug class is already available on the market. This tends to lead to lower prices for prescription medicines in general (i.e. this process also drives down the price of generic medicines as a secondary effect), and generally leads to more affordable prices of medicinal products across Europe, which can explain the low number of reports of tablet splitting practices. However, further studies are required to assess if this hypothesis is applicable on a worldwide basis.

## Conclusions

As discussed throughout this research and highlighted by other authors [e.g. 6–9, 22–25], PSPs have become an important component of the work performed by pharmaceutical companies. These programmes are not studies in the formal sense, as they are designed to provide access to medicines and/or deliver education and support to patients and healthcare professionals.

The limitations of PSPs remain manifest, and we accept that these programs are not designed to support or enhance safety signal detection [7, 8, 23, 24]. The evidence presented here shows that even within a heterogeneous data set patterns may be observed and compared in order to make sense of the observations. Indeed, the evidence here is aligned with the conclusions of at least one other research group [22] in that PSPs may play a role in improving courses of treatment, clinical outcomes, and patient satisfaction, particularly in chronic diseases such as congestive heart failure.

Based on the evidence of this project, and in common with other supporting evidence it is unlikely that PSPs will ever become a common source of medically important safety signals, newly identified medically important risks, or indeed provide evidence for the further definition of potential safety signals or the transition of potential risks to identified risks. However, PSPs can be valuable sources of real-world data to track and study patient-reported adherence or non-adherence to the medicinal product and to understand the reasons why patients take medications incorrectly. It may be due to cost as some reports showed in the sacubitril/valsartan line listing.

Moving forwards, perhaps it would worth designing some PSPs specifically to allow the collection of real-world data that could ultimately benefit the patient by supporting the evaluation of patterns of use, adherence and even pharmacoeconomic factors. The challenge to this hypothesis was that these modifications would potentially increase the complexity of PSPs and may not facilitate the fundamental aims of easing access to medicines.

Several areas would be worth conducting further research where PSPs may provide valuable data. This research revealed the pattern of tablet splitting or one daily dose with sacubitril/valsartan. Tablet splitting may lead to lack of efficacy, underdose and/or missed dose that can potentially be dangerous to the patient. A recommendation would be to conduct a drug utilisation study, specifically a voluntary PASS with a primary endpoint of efficacy to further study the effect of tablet splitting or one daily dose instead of two on efficacy of the medicinal

product. If such study would be commissioned, the MAH must carefully consider the protocol. The outcome of the study may affect the CDS and further local labelling with additional instruction no to split tablet or may prove that a lower dose would be acceptable. However, it would also be worth considering if such effort from the MAH worth it.

Information from PSPs could be crucial for pharmaceutical companies as well as regulators, indeed, there are implications for patients and HCPs. It must be highlighted that the MAH must have a clear goal when designing a PSP and ensuring that the most suitable method of data collection is in place. Beyond ICSRs and signal detection, PSPs may highlight useful information as revealed in this research. These may not be critical for label changes but there is potentially beneficial information for MAHs, patients, HCPs and even regulators, i.e. to improve public health.

A suggestion for the MAH may be to design specific PSPs that collect real-world data information and use these data for further investigation and for the generation of important evidence to support new recommendations. If this could be achieved, all stakeholders would benefit, but any transition must be carefully managed, so as not to obviate the fundamental reason for conducting PSPs, which is to improve patient access to medicines.

In conclusion, in common with other authors [e.g. 22, 25] we await further harmonised guidance on optimising PSPs. Nevertheless, it is our opinion that PSPs may yield new information, if the data collected are analysed with care. We advocate that PSPs offer a range of services that can be optimized to enable patient access to medicines, and with appropriate planning PSPs can empower patients to attain better health outcomes than in the absence of these programs.

## Supporting information

**S1 Data. Data file marketing programs Novartis _30JUN2020 v1.0 (confidential data on file).**
(XLSM)

**S1 Annex. Sacubitril/valsartan reported cases of "tablet splitting" / "once-daily dose" case numbers per country (originally).**
(DOCX)

## Acknowledgments

We would like to acknowledge the advice and guidance from Dr John Talbot at the University of Hertfordshire during the planning and conduct of this research.

## Author Contributions

**Writing – original draft:** Eszter Palffy.

**Writing – review & editing:** Eszter Palffy, David John Lewis.

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
