## [Decision Letter · Decision Letter 0]

27 Jan 2023

PONE-D-22-27481Patient Support Programmes and safety reporting: a split decision?PLOS ONE

Dear Dr. Palffy,

Thank you for submitting your manuscript to PLOS ONE. After careful consideration, we feel that it has merit but does not fully meet PLOS ONE’s publication criteria as it currently stands. Therefore, we invite you to submit a revised version of the manuscript that addresses the points raised during the review process.

The manuscript has been evaluated by two  reviewers, and their comments are available below.

The reviewers have raised a number of concerns that need attention. They request additional information on methodological aspects of the study, revisions to the statistical analyses (e.g they find that results section requires refinement and detailing of the signal data) and they note the imbalance of the discussion section. Could you please revise the manuscript to carefully address the concerns raised?

We look forward to receiving your revised manuscript.

Kind regards,

Katrien Janin

Staff Editor

PLOS ONE

Journal Requirements:

3. For studies reporting research involving human participants, PLOS ONE requires authors to confirm that this specific study was reviewed and approved by an institutional review board (ethics committee) before the study began. Please provide the specific name of the ethics committee/IRB that approved your study or explain why you did not seek approval in this case.

For additional information about PLOS ONE ethical requirements for human subject's research, please refer to http://journals.plos.org/plosone/s/submission-guidelines#loc-human-subjects-research. 

4. Please provide additional details regarding participant consent. In the ethics statement in the Methods and online submission information, please ensure that you have specified what type you obtained (for instance, written or verbal, and if verbal, how it was documented and witnessed). If your study included minors, state whether you obtained consent from parents or guardians. If the need for consent was waived by the ethics committee, please include this information.

For additional information about PLOS ONE ethical requirements for human subject's research, please refer to http://journals.plos.org/plosone/s/submission-guidelines#loc-human-subjects-research.

JS (Assoc Ed) 07/14/2022: ***Ed Office, please ping me at jstortz@plos.org before sending decision letter to authors. Added decision check flag due to the study design (company data being used for in-house company analysis).

5. In the ethics statement in the manuscript and in the online submission form, please provide additional information about the patient records/samples used in your retrospective study. Specifically, please ensure that you have discussed whether all data/samples were fully anonymized before you accessed them and/or whether the IRB or ethics committee waived the requirement for informed consent. If patients provided informed written consent to have data/samples from their medical records used in research, please include this information.

Reviewers' comments:

Reviewer's Responses to Questions

**Comments to the Author**

1. Is the manuscript technically sound, and do the data support the conclusions?

Reviewer #1: Yes

Reviewer #2: Partly

2. Has the statistical analysis been performed appropriately and rigorously? 

Reviewer #1: N/A

Reviewer #2: N/A

3. Have the authors made all data underlying the findings in their manuscript fully available?

Reviewer #1: Yes

Reviewer #2: No

4. Is the manuscript presented in an intelligible fashion and written in standard English?

Reviewer #1: Yes

Reviewer #2: Yes

5. Review Comments to the Author

Reviewer #1: Thank you very much for the opportunity to review this manuscript. The paper is well-written. The study explored an interesting topic. I have two comments/ recommendations.

In the discussion part: there is much explanation about the financial elements on saving money on medication and its relation with the pricing. I would like to encourage the author to let go the information not directly related to the objective of the manuscript.

Line 6 of the discussion; the below figure

There is no figure, please mention what is the number of the figure! And usually it is not recommended to add figures in the discussion.

For the figures, please note that each figure must have a title to be inserted below the figure.

Reviewer #2: Eszter Palffy and David John Lewis conducted a study of an important and relevant issue, which is the utility of patient support programs (PSPs) in collecting information on drug safety. In discussing the relevance and findings, the authors themselves conclude that there is an extreme paucity of published research on this issue. This is borne out by other, sadly single studies and systematic reviews published in recent years (e.g., Sacristán J. et al. (2022), Lee I. et al. (2000), Riveroll A. (preprint 2022)), among others. Unfortunately, the data from these studies have not been discussed by the authors, although they contain several important conclusions that are worth noting.

In particular, one cannot but agree that the type of PSP strongly influences the quantity and quality of safety information obtained, and that some of the programs are much more effective than others. The drug included in the PSP also plays an important role. The information obtained in oncology will be quite different from that obtained when a cardiologic or gastroenterologic drug is used. These considerations could be discussed in the manuscript.

The authors further point out that only two new safety signals were generated during the PSPs of sacubitril/valsartan, despite the large number of reports received. However, a safety signal, according to GVP definition, is not only new risks, but also renewed information about risks already known, their frequency, outcomes, severity, etc. It is not clear from the results presented whether or not the PSP data contributed to a change in knowledge about the previously known risks of sacubutril/valsartan.

It is known that analysis of validated signals can lead not only to risk confirmation or determination of the need for further analysis, but also to signal rejection. It would be important for the reader to know whether such, refuted signals were obtained in the PSP presented. Thus, in my opinion, the results section requires refinement and detailing of the signal data.

An indication of the signal detection methodology used in the company is also important. It is known that the sensitivity of various methods for detecting the disproportionality of reporting is quite different and signals detected using one method can be missed when using another. Also of importance is the threshold of signal detection defined in the company. These features are fairly well understood (Dijkstra et al., 2020) and are presented in relevant manuals such as SCOPE WP5.

Much of the manuscript is devoted to findings in the use of sacubutril/valsartan ("tablet splitting"/"once-daily dose") and an analysis of such risky practices by patients in different countries, including an analysis of the economic rationale. In this regard, we should note the imbalance of the discussion section. Paying considerable attention to the analysis of economic and social reasons for the problem of pill splitting and dosage regimen violations, the authors missed the discussion of the consequences of such violations from the perspective of pharmacotherapy safety for patients (risk of overdose, development of type A ADR or ineffectiveness, etc.). In connection with the mentioned above, it would be appropriate to analyze the relationship between the newly revealed risks and the uncovered dose irregularities.

Among the minor findings the lack of appropriate explanations for the tables presented (e.g., color differentiation of rows), the meaning of Figure 2 in the part devoted to sources of information other than PSPs should be noted. A more detailed title of the article should also be considered.

In this connection, the reviewed manuscript is of practical interest, but needs to be improved in the part of the proposed comments.

6. PLOS authors have the option to publish the peer review history of their article (what does this mean?). If published, this will include your full peer review and any attached files.

Reviewer #1: No

Reviewer #2: **Yes: **Aleksandr Matveev

---

## [Author Response · Author response to Decision Letter 0]

28 Jun 2023

Response: Manuscript amended accordingly. 

Response: Correction made to add a title page, with details as required.

3. For studies reporting research involving human participants, PLOS ONE requires authors to confirm that this specific study was reviewed and approved by an institutional review board (ethics committee) before the study began. Please provide the specific name of the ethics committee/IRB that approved your study or explain why you did not seek approval in this case.

Response: Please refer to Section 9 of the attached Master Thesis Proposal, ‘Ethical consideration for human subjects’. There was no requirement for review by an ethics committee, as this was a retrospective review of consented safety data from patient support programmes. 

For additional information about PLOS ONE ethical requirements for human subject's research, please refer to http://journals.plos.org/plosone/s/submission-guidelines#loc-human-subjects-research. 

4. Please provide additional details regarding participant consent. In the ethics statement in the Methods and online submission information, please ensure that you have specified what type you obtained (for instance, written or verbal, and if verbal, how it was documented and witnessed). If your study included minors, state whether you obtained consent from parents or guardians. If the need for consent was waived by the ethics committee, please include this information. 

Response: Please refer to our response to point 3 and the Master Thesis Proposal that was accepted by the University of Hertfordshire review board. 

For additional information about PLOS ONE ethical requirements for human subject's research, please refer to http://journals.plos.org/plosone/s/submission-guidelines#loc-human-subjects-research.

JS (Assoc Ed) 07/14/2022: ***Ed Office, please ping me at jstortz@plos.org before sending decision letter to authors. Added decision check flag due to the study design (company data being used for in-house company analysis).

5. In the ethics statement in the manuscript and in the online submission form, please provide additional information about the patient records/samples used in your retrospective study. Specifically, please ensure that you have discussed whether all data/samples were fully anonymized before you accessed them and/or whether the IRB or ethics committee waived the requirement for informed consent. If patients provided informed written consent to have data/samples from their medical records used in research, please include this information. 

Response: Please refer to our responses to points 3 and 4.We have ensured the protection of data by pseudonymisation. Neither of the data reviewers had access to any personally identifiable information for an individual patient throughout the study. The raw data provided shows the content that was reviewed, with each report being identified by a unique primary key allocated in a random order.

Response: Data availability: The dataset for this manuscript is not publicly available because of the data protection policy of the sources that provided data for this study. Requests to access the dataset should be directed to the first author, and access will be granted on reasonable request. Additionally, the data used during this research was a cohort line listing from the database and no personally identifiable data were viewed or used during the research. 

Response: The line listing retrieved from the private database that was used for the research is provided as an attachment to our response. 

Response: We request that use the data availability statement below:

“The datasets for this manuscript are not publicly available because of the data protection policy of the sources that provided data for this study. Requests to access the datasets should be directed to the first author, and access will be granted on reasonable request.”

Response: Ethics statement is now included under the “Methods” section.

 

Reviewers' comments:

Reviewer's Responses to Questions

Comments to the Author

1. Is the manuscript technically sound, and do the data support the conclusions?

Reviewer #1: Yes

Reviewer #2: Partly

Response: We hope that our proposed changes will address the comment from Reviewer 2.

2. Has the statistical analysis been performed appropriately and rigorously? 

Reviewer #1: N/A

Reviewer #2: N/A

3. Have the authors made all data underlying the findings in their manuscript fully available?

Reviewer #1: Yes

Reviewer #2: No 

Response: We have provided the raw data output that was used as the basis of this research. No personally identifiable information (PII) is present in the output, as PII was not required for the purpose of this research.

4. Is the manuscript presented in an intelligible fashion and written in standard English?

Reviewer #1: Yes

Reviewer #2: Yes

5. Review Comments to the Author

Reviewer #1: Thank you very much for the opportunity to review this manuscript. The paper is well-written. The study explored an interesting topic. I have two comments/ recommendations.

In the discussion part: there is much explanation about the financial elements on saving money on medication and its relation with the pricing. I would like to encourage the author to let go the information not directly related to the objective of the manuscript.

Response: Thank you for this advice. We have reviewed the manuscript and removed sections of text that were not directly relevant to the objective.

Line 6 of the discussion; the below figure 

There is no figure, please mention what is the number of the figure! And usually it is not recommended to add figures in the discussion.

For the figures, please note that each figure must have a title to be inserted below the figure.

Response: We apologise for inadvertent omission of the figure. This has been added in the correct place. 

Reviewer #2: Eszter Palffy and David John Lewis conducted a study of an important and relevant issue, which is the utility of patient support programs (PSPs) in collecting information on drug safety. In discussing the relevance and findings, the authors themselves conclude that there is an extreme paucity of published research on this issue. This is borne out by other, sadly single studies and systematic reviews published in recent years (e.g., Sacristán J. et al. (2022), Lee I. et al. (2000), Riveroll A. (preprint 2022)), among others. Unfortunately, the data from these studies have not been discussed by the authors, although they contain several important conclusions that are worth noting.

In particular, one cannot but agree that the type of PSP strongly influences the quantity and quality of safety information obtained, and that some of the programs are much more effective than others. The drug included in the PSP also plays an important role. The information obtained in oncology will be quite different from that obtained when a cardiologic or gastroenterologic drug is used. These considerations could be discussed in the manuscript. 

Response: We first drafted our manuscript before the publication of these papers, hence we appreciate the advice to add more up-to-date citations. After conducting a further literature review we have made reference to the publications below in our discussion and conclusions:

1. Sacristán JA, Artime E, Díaz-Cerezo S, Comellas M, Pérez-Carbonell L, Lizán L. The Impact of Patient Support Programs in Europe: A Systematic Literature Review. Patient. 2022 Nov;15(6):641-654. doi: 10.1007/s40271-022-00582-y. Epub 2022 Jun 21. PMID: 35725866; PMCID: PMC9584873.

2. Lee I, Lee TA, Crawford SY, Kilpatrick RD, Calip GS, Jokinen JD. Impact of adverse event reports from marketing authorization holder-sponsored patient support programs on the performance of signal detection in pharmacovigilance. Expert Opin Drug Saf. 2020 Oct;19(10):1357-1366. doi: 10.1080/14740338.2020.1792883. Epub 2020 Jul 27. PMID: 32662668.

3. Lee I, Jokinen JD, Crawford SY, Calip GS, Kilpatrick RD, Lee TA. Exploring Completeness of Adverse Event Reports as a Tool for Signal Detection in Pharmacovigilance. Ther Innov Regul Sci. 2021 Jan;55(1):142-151. doi: 10.1007/s43441-020-00199-z. Epub 2020 Jul 27. PMID: 32720297.

4. Riveroll AL et al. A Scoping Review of Patient Support Program Services Across Diverse Settings and Disease Areas Described from a People-Centered Perspective. Preprint 2022 https://doi.org/10.21203/rs.3.rs-1447984/v1.

The authors further point out that only two new safety signals were generated during the PSPs of sacubitril/valsartan, despite the large number of reports received. However, a safety signal, according to GVP definition, is not only new risks, but also renewed information about risks already known, their frequency, outcomes, severity, etc. It is not clear from the results presented whether or not the PSP data contributed to a change in knowledge about the previously known risks of sacubutril/valsartan. It is known that analysis of validated signals can lead not only to risk confirmation or determination of the need for further analysis, but also to signal rejection. It would be important for the reader to know whether such, refuted signals were obtained in the PSP presented. Thus, in my opinion, the results section requires refinement and detailing of the signal data.

Response: We have provided the additional text below, and a diagram to clarify the signal management process that was applied. Three different signal detection techniques were applied.

We confirm that no change in the documented safety profile of sacubitril/valsartan was detected as a result of the safety data generated from PSPs. Signal detection and signal management was conducted using disproportionality statistics calculated according to the method detailed in Berlin et al (2012). Observations of disproportional statistics undergo medical and scientific review according to the signal management process in the diagram below:

In addition to routine use of the disproportionality tool we review designated medical events

(DME) and assess signals of increased frequency, noting trends over time increased severity measured by proxy as increased fatal or hospitalisations. At no stage did any of the safety data from PSPs lead to a change in the documented safety profile of sacubitril/valsartan. Consequently there were no changes to the core safety information for the product, and no safety variations were submitted after a thorough evaluation of the safety data from the PSPs.

An indication of the signal detection methodology used in the company is also important. It is known that the sensitivity of various methods for detecting the disproportionality of reporting is quite different and signals detected using one method can be missed when using another. Also of importance is the threshold of signal detection defined in the company. These features are fairly well understood (Dijkstra et al., 2020) and are presented in relevant manuals such as SCOPE WP5. 

Response: We have added text, and provided a diagram of the signal management process that was used (Figure 3).

Much of the manuscript is devoted to findings in the use of sacubutril/valsartan ("tablet splitting"/"once-daily dose") and an analysis of such risky practices by patients in different countries, including an analysis of the economic rationale. In this regard, we should note the imbalance of the discussion section. Paying considerable attention to the analysis of economic and social reasons for the problem of pill splitting and dosage regimen violations, the authors missed the discussion of the consequences of such violations from the perspective of pharmacotherapy safety for patients (risk of overdose, development of type A ADR or ineffectiveness, etc.). In connection with the mentioned above, it would be appropriate to analyze the relationship between the newly revealed risks and the uncovered dose irregularities. 

Response: We agree with this suggestion, and we have carefully considered the risk of underdosing and/or missed doses leading to lack of efficacy. We have included comments in the form of a further recommendation in our conclusions towards the end of manuscript. It is important to recognise that the prescribing information for the product includes the recommendation: ‘If a dose is missed, the patient should take the next dose at the scheduled time. Splitting or crushing of the tablets is not recommended.’

Among the minor findings the lack of appropriate explanations for the tables presented (e.g., colour differentiation of rows), the meaning of Figure 2 in the part devoted to sources of information other than PSPs should be noted. A more detailed title of the article should also be considered. 

Response: We accept these suggestions, and hope that our added explanatory text, new figure (Fig. 3), and the revised text has provided the additional clarity required.

In this connection, the reviewed manuscript is of practical interest, but needs to be improved in the part of the proposed comments.

6. PLOS authors have the option to publish the peer review history of their article (what does this mean?). If published, this will include your full peer review and any attached files.

Reviewer #1: No. Anonymous

Reviewer #2: Yes: Aleksandr Matveev

---

## [Decision Letter · Decision Letter 1]

26 Sep 2023

PONE-D-22-27481R1Patient Support Programmes and safety reporting: a split decision?PLOS ONE

Dear Dr. Palffy,

Thank you for submitting your manuscript to PLOS ONE. After careful consideration, we feel that it has merit but does not fully meet PLOS ONE’s publication criteria as it currently stands. Therefore, we invite you to submit a revised version of the manuscript that addresses the points raised during the review process.

We look forward to receiving your revised manuscript.

Kind regards,

Masoud Behzadifar

Academic Editor

PLOS ONE

Journal Requirements:

Reviewers' comments:

Reviewer's Responses to Questions

**Comments to the Author**

1. If the authors have adequately addressed your comments raised in a previous round of review and you feel that this manuscript is now acceptable for publication, you may indicate that here to bypass the “Comments to the Author” section, enter your conflict of interest statement in the “Confidential to Editor” section, and submit your "Accept" recommendation.

Reviewer #1: All comments have been addressed

Reviewer #3: (No Response)

Reviewer #4: All comments have been addressed

Reviewer #5: All comments have been addressed

2. Is the manuscript technically sound, and do the data support the conclusions?

Reviewer #1: Yes

Reviewer #3: (No Response)

Reviewer #4: Yes

Reviewer #5: Yes

3. Has the statistical analysis been performed appropriately and rigorously? 

Reviewer #1: N/A

Reviewer #3: (No Response)

Reviewer #4: Yes

Reviewer #5: I Don't Know

4. Have the authors made all data underlying the findings in their manuscript fully available?

Reviewer #1: Yes

Reviewer #3: (No Response)

Reviewer #4: Yes

Reviewer #5: Yes

5. Is the manuscript presented in an intelligible fashion and written in standard English?

Reviewer #1: Yes

Reviewer #3: (No Response)

Reviewer #4: Yes

Reviewer #5: Yes

6. Review Comments to the Author

Reviewer #1: All the comments presented in the first round were addressed. For the first comment, the author was able to satisfy the requested explanation. For the second comment, the author noted the need to address the information requested as well.

Reviewer #3: • Title of study is not informative. You should indicate the study design and location in title.

• In abstract, you should clarify your main goal.

• Introduction is too complex. Please concise it.

• In introduction, you should specify current gaps and clarify your main goals.

• You should explain your methods in more details so that any researcher be able to repeat it.

• Table 4 is not necessary. You can insert such information in the text.

• In the first paragraph of discussion, you should report your main findings.

• Your conclusion is too long.

Reviewer #4: I believe this work deserves to be published and would greatly contribute to the scholarly community.

Reviewer #5: Dear Editor

I read the manuscript revisioning file. The authors have responded the comments correctly and this format is ready for publishing.

7. PLOS authors have the option to publish the peer review history of their article (what does this mean?). If published, this will include your full peer review and any attached files.

Reviewer #1: No

Reviewer #3: No

Reviewer #4: No

Reviewer #5: No

---

## [Author Response · Author response to Decision Letter 1]

6 Nov 2023

We believe that Reviewers #1, #4 and #5 were satisfied with the content of the manuscript and we have addressed the points made by Reviewer #3.

• Title of study is not informative. You should indicate the study design and location in title.

A. We have proposed a new title – see revision.

• In abstract, you should clarify your main goal.

A. Main goal clarified in paragraph 2 of the abstract.

• Introduction is too complex. Please concise it.

A. Four reviewers have supported the current text; one has raised comments. As this is a complex topic, we prefer to preserve the majority of our introduction. Nevertheless, we have reduced the text where possible.

• In introduction, you should specify current gaps and clarify your main goals.

A. Gaps are highlighted within the new text on the inspection history. The main goals of our research have been clarified in the abstract.

• You should explain your methods in more details so that any researcher be able to repeat it.

A. We have described in the Materials section the products, the programmes, the existing reference documentation, and the evidence gathered, as well as the processes and the POPSys database. Within the Methods section, we have described our research. In our view this research may be repeated by others who have access to data from PSPs. 

• Table 4 is not necessary. You can insert such information in the text.

A. We wish to request that Table 4 is placed as an annex to the paper. We have added text to allow us to remove this table into an annex, so that key information is visible to readers, thus we feel that we have provided an appropriate response to this comment.

• In the first paragraph of discussion, you should report your main findings.

A. Paragraph moved to clarify main (unexpected) finding at the start of the Discussion.

• Your conclusion is too long.

A. We were asked to add three paragraphs to the Conclusion by three other reviewers, hence we are reluctant to remove text at this stage, as it would displease the majority of the reviewers. The topic requires care, such that we do not dismiss real world evidence provided directly by patients. Nevertheless, we have reduced the text where possible.

---

## [Decision Letter · Decision Letter 2]

20 Nov 2023

Real-World Evidence Revelations: The Potential of Patient Support Programmes to Provide Data on Medication Usage

PONE-D-22-27481R2

Dear Dr. Palffy,

We’re pleased to inform you that your manuscript has been judged scientifically suitable for publication and will be formally accepted for publication once it meets all outstanding technical requirements.

Kind regards,

Masoud Behzadifar

Academic Editor

PLOS ONE

Additional Editor Comments (optional):

Reviewers' comments:

Reviewer's Responses to Questions

**Comments to the Author**

1. If the authors have adequately addressed your comments raised in a previous round of review and you feel that this manuscript is now acceptable for publication, you may indicate that here to bypass the “Comments to the Author” section, enter your conflict of interest statement in the “Confidential to Editor” section, and submit your "Accept" recommendation.

Reviewer #4: All comments have been addressed

Reviewer #5: All comments have been addressed

2. Is the manuscript technically sound, and do the data support the conclusions?

Reviewer #4: Partly

Reviewer #5: Yes

3. Has the statistical analysis been performed appropriately and rigorously? 

Reviewer #4: N/A

Reviewer #5: Yes

4. Have the authors made all data underlying the findings in their manuscript fully available?

Reviewer #4: Yes

Reviewer #5: Yes

5. Is the manuscript presented in an intelligible fashion and written in standard English?

Reviewer #4: Yes

Reviewer #5: Yes

6. Review Comments to the Author

Reviewer #4: (No Response)

Reviewer #5: I believe the authors have adequately addressed my comments and the paper is suitable for publication.

7. PLOS authors have the option to publish the peer review history of their article (what does this mean?). If published, this will include your full peer review and any attached files.

Reviewer #4: No

Reviewer #5: **Yes: **Dr Hasan Abolghasem Gorji

---

## [Editor Report · Acceptance letter]

30 Jan 2024

PONE-D-22-27481R2 

PLOS ONE

Dear Dr. Palffy, 

I'm pleased to inform you that your manuscript has been deemed suitable for publication in PLOS ONE. Congratulations! Your manuscript is now being handed over to our production team.

Kind regards, 

on behalf of

Dr. Masoud Behzadifar 

Academic Editor

PLOS ONE